

# Genetic polymorphisms in the *IFNL4*, *MxA*, and *MxB* genes were associated with biochemical index of chronic HBV patients from Yunnan, China

Kexi Zheng[1], Yunsong Shen[2], Xueshan Xia[1], Yuzhu Song[1] and
A-Mei Zhang[1]

[1] Kunming University of Science and Technology, Kunming, China
[2] Kunming Angel Women's & Children's Hospital, Kunming, China

Corresponding author
A-Mei Zhang, zam1980@yeah.net

## ABSTRACT

Hepatitis B virus (HBV) infection causes Hepatitis B, which is one of the most common causes of hepatocellular carcinoma (HCC). The single nucleotide polymorphisms (SNPs) of the host immune genes could impact HBV infection, viral clearance, and treatment effect. However, the contradictory roles of several studies suggest further analysis of various populations. The whole blood and biochemical indexes of 448 HBV patients and matched controls were collected from the Yunnan population to investigate the genetic roles of *IFNL4* and the downstream genes (*MxA* and *MxB*). The genotypes, alleles, and haplotypes frequencies of the seven SNPs (rs11322783, rs117648444, rs2071430, rs17000900, rs9982944, rs408825, and rs2838029) from the HBV patients and controls were analyzed. However, no association was identified between the SNPs and HBV infection. Then, biochemical index levels were evaluated among the HBV patients with different genotypes of the seven SNPs. The results indicated that the liver function index levels (including alanine transaminase (ALT), aspartate transaminase (AST), total bilirubin (TBIL), direct bilirubin (DBIL), indirect bilirubin (IBIL), and albumin (ALB)) were influenced by the genotypes of the SNPs in HBV patients. Moreover, when the HBV patients were divided into HBsAg-positive and -negative groups, the association between the SNP genotypes and the biochemical indexes still existed. In addition, although the genetic polymorphisms in the *IFNL4*, *MxA*, and *MxB* genes were not significantly associated with HBV infection in the Yunnan population, these genes could indirectly influence disease progression by associating with the biochemical index levels of Yunnan HBV patients.

## INTRODUCTION

The Hepatitis B virus (HBV) was identified by Dr. Baruch Blumberg in 1964, and was later characterized as the cause of hepatitis B disease. It is one of the most popular reasons for serious hepatitis diseases (*Glebe et al., 2021*). In adults, 5% of HBV-infected persons progress into chronic HBV (CHB) infection, and 20–30% of CHB could develop cirrhosis

and hepatocellular carcinoma (HCC). According to the World Health Organization (WHO) reports, 275 million individuals (about 3.5% of the population) live with CHB. The epidemic regions of CHB are primarily located in the Western Pacific Region and Africa (*WHO, 2017*). Although vaccine usage has significantly protected individuals from HBV infection, the number of adult HBV patients is still significant.

Interferons (IFNs) are commonly used for HBV therapy in clinics, and IFN-induced immune pathways play an essential role in CHB infection by activating the interferon-stimulated gene (ISG) expression. In addition, IFNs could stimulate the JAK pathway and regulate STAT1 transcription (*Mani & Andrisani, 2019*). However, the treatment effects revealed significant differences among HBV patients, and the immunological genetic factors in the host are necessary for antiviral response (*Brouwer et al., 2019*). Distinguishing from other members of the IFN family, IFNL4 could impair hepatitis C virus (HCV) clearance (*Prokunina-Olsson et al., 2013*). The viral clearance of HCV patients treated with pegylated-IFN-α/ribavirin could be influenced by the genetic polymorphisms of the *IFNL4* gene (*O'Brien et al., 2015*). Similarly, the genetic polymorphisms in the *IFNL4* gene were also associated with the viral load of HBV patients and protect individuals from HBV infection (*Chihab et al., 2021*; *Grzegorzewska et al., 2020*).

The *MX dynamin-like GTPase* genes, including the *MxA* gene (or *Mx1*) and the *MxB* gene (or *Mx2*), belonged to the IFN-stimulated gene (ISG). The amino acid homology between *MxA* and *MxB* reached 63%, but their antiviral activities differed significantly. Although *MxA* is a well-known wide-spectrum antiviral factor (*Haller et al., 2015*), *MxA* and *MxB* could inhibit HBV replication (*Wang et al., 2020*). However, whether the genetic polymorphisms of the *MX* genes could influence HBV infection and pathogenesis has not been adequately studied.

The present study analyzed the genotype, allele, and haplotype frequencies of genetic polymorphisms of the *IFNL4*, *MxA*, and *MxB* genes in HBV patients from Yunnan, China. In addition, the association between biochemical indexes and genotypes of the genetic polymorphisms was identified.

## MATERIALS AND METHODS

### Individuals and biochemical index

A total of 448 HBV infected persons were identified as CHB patients without any treatment, and medical doctors also recruited 448 general controls from the First People's Hospital of Yunnan Province between 2017 and 2020. Data were collected based on our previous description in *Song et al. (2017)*. Specifically, all the patients infected with HBV but without HCV and Human Immunodeficiency virus (HIV) infection were diagnosed as CHB patients based on the clinical phenotype and biochemical indexes provided by medical doctors. Serology indexes of HBV patients were determined by the Quantitative CLIA Kit (Autobio, Zhengzhou, China) according to the manual. The controls were not infected by any virus and were without any disease, and were recruited from normal individuals after physical examination from the hospital. A total of

3 mL of whole blood was obtained from each individual to extract the genomic DNA (gDNA) for analyzing single nucleotide polymorphisms (SNPs) (*Song et al., 2017*).

The biochemical information was obtained from each participant, including alanine transaminase (ALT), aspartate transaminase (AST), total bilirubin (TBIL), direct bilirubin (DBIL), indirect bilirubin (IBIL), total protein (TP), albumin (ALB), globin (GLOB), blood urea nitrogen (BUN), serum creatinine (CREA), serum uric acid (UA), blood glucose (GLU), white blood cells (WBC), neutrophilic granulocyte (NEUT), lymphocytes (LYM), monocytes (MONO), eosinophil granulocyte (EO), and basophile granulocyte (BASO). In addition, the Hepatitis B surface antigen (HBsAg) was positive or negative in each HBV sample.

Written informed consent was obtained from each participant, conforming to the tenets of the Declaration of Helsinki before the study. The Institutional Review Board of Kunming University of Science and Technology approved the present study (Approval No. 2014SK027).

### SNP genotyping and haplotype construction

Two SNPs (rs11322783 and rs117648444) in the *IFNL4* gene, two SNPs (rs2071430 and rs17000900) in the *MxA* gene, and three SNPs (rs9982944, rs408825, and rs2838029) in the *MxB* gene were genotyped by using the SnapShot method (ABI, Los Angeles, CA, USA) (*Ben-Avi et al., 2004*). First, the amplifying and extending primers were designed for each SNP. Then, PCR reactions were performed with two kinds of primers that were terminated behind one base at the end of extending primers. Finally, genotypes for each SNP were determined using the fluorescence color. Seven SNPs were tag SNPs, and the collection principle was that the minor allele frequency of these SNPs should be more than 2%.

Haplotypes were constructed using seven SNPs from HBV patients and controls in the SHEsis software platform (http://analysis.bio-x.cn/myAnalysis.php). The lowest frequency threshold (LFT) for haplotype analysis was 0.05. In addition, linkage disequilibrium (LD) was calculated among the seven SNPs (*Li et al., 2009*).

### Data analysis

The data from each biochemical index was represented as mean ± SEM in HBV patients and controls. The biochemical indexes between HBV patients and controls were compared using the Student's *t* test (two-tailed). The frequencies of genotype, allele, and haplotypes were compared between the HBV patients and controls using the chi-square test with Yates' correction. The association between genotypes and biochemical indexes was analyzed by using Student's *t* test (two-tailed). Genotype and allele frequencies of each SNP were compared between the HBV patients with HBsAg-positive and -negative. A *P*-value less than 0.05 was considered statistically significance.

## RESULTS

### Basic information

The mean age of the HBV patients and the controls were 42.12 ± 0.38 and 40.58 ± 0.53 years, respectively. Although the number of males was more in controls (*N* = 275, 61.38%)

**Table 1 Analysis of biochemical index between HBV infected persons and controls.**

| | HBV patients | Controls | P-value |
|---|---|---|---|
| Gender | | | |
| Male (%) | 245 (54.69%) | 275 (61.38%) | >0.05 |
| Female (%) | 203 (45.31%) | 173 (38.62%) | >0.05 |
| Age | 42.12 ± 0.38 | 40.58 ± 0.53 | >0.05 |
| AST (U/L) | 37.37 ± 4.07 | 24.63 ± 0.53 | 0.002 |
| ALT (U/L) | 47.49 ± 7.30 | 29.00 ± 1.04 | 0.013 |
| TBIL (μmol/L) | 14.78 ± 0.89 | 11.66 ± 0.28 | 0.0009 |
| DBIL (μmol/L) | 6.40 ± 0.69 | 3.86 ± 0.09 | 0.0003 |
| IBIL (μmol/L) | 8.39 ± 0.25 | 7.78 ± 0.19 | 0.053 |
| TP (g/L) | 73.12 ± 0.37 | 78.57 ± 0.20 | <0.0001 |
| ALB (g/L) | 40.78 ± 0.29 | 47.18 ± 0.13 | <0.0001 |
| GLOB (g/L) | 32.25 ± 0.26 | 31.53 ± 0.18 | 0.010 |
| GLU (mmol/L) | 4.88 ± 0.07 | 5.34 ± 0.06 | <0.0001 |
| BUN (mmol/L) | 4.94 ± 0.13 | 5.04 ± 0.06 | 0.491 |
| CREA (μmol/L) | 75.86 ± 4.22 | 71.56 ± 0.79 | 0.317 |
| UA (μmol/L) | 344.6 ± 4.91 | 352.1 ± 4.59 | 0.265 |
| WBC ($10^9$/L) | 9.28 ± 2.42 | 6.70 ± 0.12 | 0.288 |
| NEUC ($10^9$/L) | 4.41 ± 0.13 | 3.77 ± 0.06 | <0.0001 |
| LYM ($10^9$/L) | 1.82 ± 0.03 | 2.25 ± 0.03 | <0.0001 |
| MONO ($10^9$/L) | 0.43 ± 0.01 | 0.57 ± 0.11 | 0.199 |
| EO ($10^9$/L) | 0.15 ± 0.01 | 0.14 ± 0.005 | 0.361 |
| BASO ($10^9$/L) | 0.03 ± 0.001 | 0.03 ± 0.001 | 0.0005 |

than that in HBV patients ($N$ = 245, 54.69%), it showed no significance. Male to female ratio was 1.2:1 and 1.6:1 in HBV patients and controls, respectively. Other biochemical indexes showed significant differences between HBV patients and controls after excluding IBIL, BUN, CREA, UA, WBC, MONO, and EO (Table 1). These results suggested that liver function was impaired in HBV patients.

## No association between SNPs in three genes and HBV infection

Genotype and allele frequencies of the seven SNPs revealed no significant difference between the HBV patients and controls (Table 2). No individual was identified to carry genotype AA of rs117648444. The D' value between the SNPs rs2071430 and rs17000900 in the *MxA* gene and the SNPs rs408825 and rs2838029 in the *MxB* gene were 0.96 and 0.97, respectively. However, the $r^2$ value had no linkage disequilibrium among these SNPs (Fig. 1). Although the seven SNPs were tagSNPs, they could not tag each other. Then, 31 and 35 haplotypes were constructed in HBV patients and controls. After analyzing the seven haplotypes which frequencies were more than 5%, no haplotype showed a statistical difference between the HBV patients and the controls (Table 3).
**Table 2  Analysis of genotypes and alleles in the *IFNL4*, *MxA*, and *MxB* genes between HBV infected persons and controls.**

| SNP | HBV patients (N = 448) | Controls (N = 448) | P-value | OR (95% CI) |
|---|---|---|---|---|
| rs11322783 (*IFNL4*) | | | | |
| Genotype | | | | |
| ΔG | 1 | 1 | 0.479 | 1.000 [0.053–19.04] |
| ΔG/T | 40 | 25 | 0.071 | 1.659 [1.000–2.812] |
| TT | 407 | 422 | 0.075 | 0.612 [0.368–1.000] |
| Allele | | | | |
| ΔG | 42 | 27 | 0.086 | 1.583 [0.970–2.566] |
| T | 854 | 869 | | 0.632 [0.390–1.031] |
| rs117648444 (*IFNL4*) | | | | |
| Genotype | | | | |
| AA | 0 | 0 | – | – |
| AG | 11 | 7 | 0.475 | 1.586 [0.627–4.119] |
| GG | 437 | 441 | 0.475 | 0.631 [0.243–1.595] |
| Allele | | | | |
| A | 11 | 7 | 0.477 | 1.579 [0.630–4.115] |
| G | 885 | 889 | | 0.634 [0.243–1.588] |
| rs2071430 (*MxA*) | | | | |
| Genotype | | | | |
| GG | 230 | 217 | 0.423 | 1.123 [0.867–1.456] |
| GT | 168 | 188 | 0.195 | 0.830 [0.632–1.087] |
| TT | 50 | 43 | 0.511 | 1.183 [0.767–1.801] |
| Allele | | | | |
| G | 628 | 622 | 0.797 | 1.032 [0.844–1.262] |
| T | 268 | 274 | | 0.969 [0.792–1.185] |
| rs17000900 (*MxA*) | | | | |
| Genotype | | | | |
| AA | 12 | 13 | 0.999 | 0.921 [0.414–1.988] |
| AC | 112 | 114 | 0.939 | 0.977 [0.726–1.314] |
| CC | 324 | 321 | 0.882 | 1.034 [0.775–1.380] |
| Allele | | | | |
| A | 136 | 140 | 0.844 | 0.966 [0.746–1.250] |
| C | 760 | 756 | | 1.035 [0.800–1.340] |
| rs9982944 (*MxB*) | | | | |
| Genotype | | | | |
| AA | 37 | 40 | 0.812 | 0.918 [0.576–1.452] |
| AG | 204 | 222 | 0.255 | 0.851 [0.656–1.103] |
| GG | 207 | 186 | 0.178 | 1.210 [0.930–1.576] |
| Allele | | | | |
| A | 278 | 302 | 0.246 | 0.885 [0.725–1.079] |
| G | 618 | 594 | | 1.130 [0.926–1.380] |

(Continued)
| SNP | HBV patients (N = 448) | Controls (N = 448) | P-value | OR (95% CI) |
|---|---|---|---|---|
| **Table 2 (continued)** | | | | |
| rs408825 (*MxB*) | | | | |
| Genotype | | | | |
| CC | 19 | 18 | 0.999 | 1.058 [0.560–2.024] |
| CT | 126 | 131 | 0.768 | 0.947 [0.712–1.260] |
| TT | 303 | 299 | 0.831 | 1.041 [0.784–1.370] |
| Allele | | | | |
| C | 164 | 167 | 0.903 | 0.978 [0.771–1.240] |
| T | 732 | 729 | | 1.022 [0.806–1.297] |
| rs2838029 (*MxB*) | | | | |
| Genotype | | | | |
| AA | 1 | 5 | 0.219 | 0.198 [0.017–1.434] |
| AG | 60 | 58 | 0.921 | 1.040 [0.702–1.544] |
| GG | 387 | 385 | 0.923 | 1.038 [0.712–1.516] |
| Allele | | | | |
| A | 62 | 68 | 0.649 | 0.905 [0.635–1.284] |
| G | 834 | 828 | | 1.105 [0.779–1.576] |

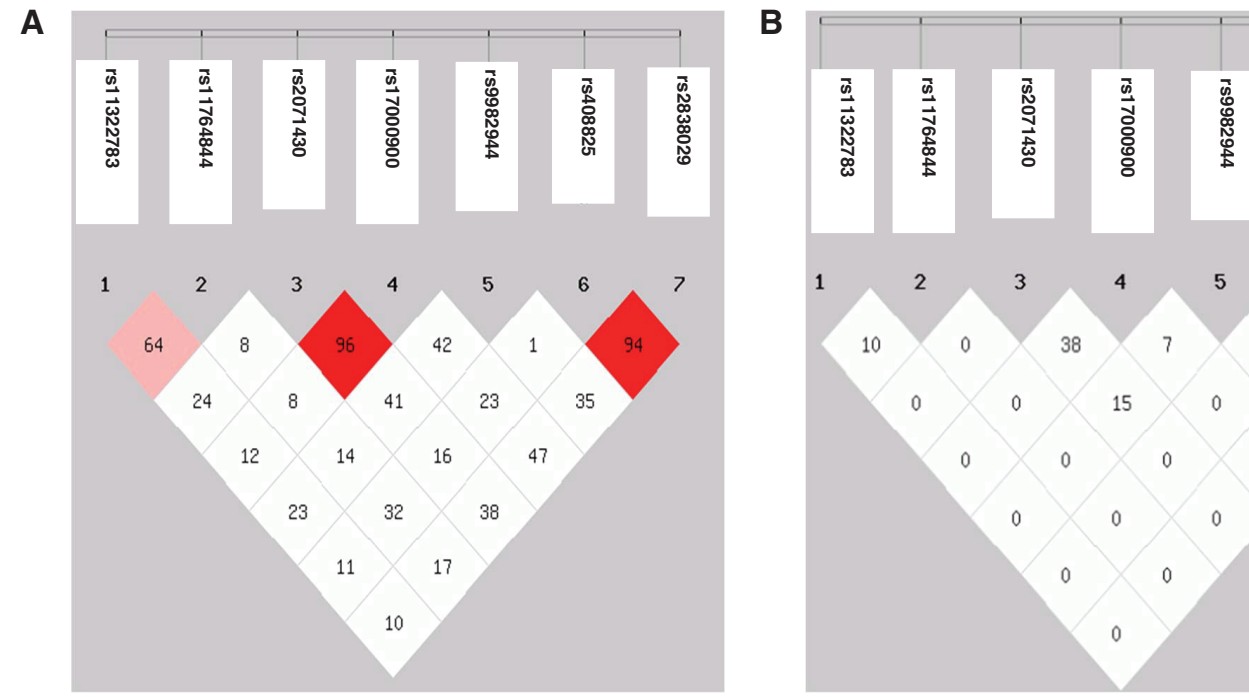

**Figure 1 Linkage disequilibrium map of seven SNPs in three genes.** (A) D′ value; (B) r² value.

**Table 3 Haplotype analysis constructed by seven SNPs between HBV infected persons and controls.**

| Haplotype | HBV patients | Controls | P-value | OR (95% CI) |
|---|---|---|---|---|
| TGGCATG | 85 | 97 | 0.386 | 0.872 [0.639–1.189] |
| TGGCGCA | 42 | 54 | 0.238 | 0.778 [0.512–1.182] |
| TGGCGTG | 405 | 378 | 0.094 | 1.189 [0.971–1.456] |
| TGTAATG | 68 | 66 | 0.822 | 1.042 [0.730–1.486] |
| TGTAGTG | 34 | 51 | 0.065 | 0.660 [0.423–1.030] |
| TGTCATG | 63 | 72 | 0.456 | 0.874 [0.612–1.246] |
| TGTCGTG | 49 | 38 | 0.184 | 1.344 [0.868–2.081] |
| Others | 150 | 140 | – | – |

## Biochemical indexes showed significant differences among HBV patients

Genotypes of six SNPs were associated with the biochemical indexes of HBV patients. The genotype ΔG of rs11322783 and genotype AA in rs2838029 existed in only one HBV patient and one control person. As a result, the genotype ΔG was combined with genotype ΔG/T of rs11322783, and genotype AA was combined with genotype AG in rs2838029 for further analysis. Biochemical indexes were significantly different among HBV patients with various genotypes of each SNP (Fig. 2). In the *IFNL4* gene, LYM ($P = 0.007$) and MONO ($P = 0.010$) levels were significantly higher in patients with genotype TT than those with genotype GG and GT of rs11322783. Genotype TT of rs2071430 seemed to reduce ALT and CREA levels in patients. The ALT, AST, and CREA levels were statistically lower in patients with genotype AA of rs17000900. The AST, TBIL, DBIL, IBIL, and MONO levels were higher in patients with genotype GG of rs2838029 than in patients with the other two genotypes. Genotype TT of rs408825 could be a risk factor for DBIL, IBIL, and EO levels in Yunnan HBV patients. ALB level were significantly higher in patients with genotype AA of rs9982944 than in other patients.

## Rs17000900 associated with HBsAg expression and biochemical indexes

HBsAg was a bio-marker for HBV cccDNA replication in the clinic. Thus, the genotype and allele frequencies were compared between HBV patients with HBsAg-positive and HBsAg-negative diagnose (Table 4). The results showed that the frequency of genotype AC in rs17000900 was significantly higher in the HBsAg-negative patients (69/238, 28.99%) than in the HBsAg-positive patients (43/210, 20.48%).

Biochemical indexes also expressed significant differences between the HBsAg-positive and the HBsAg-negative HBV patients (Fig. 3). The AST and TBIL level was higher in HBsAg-positive patients than in HBsAg-negative patients with genotype AC of rs17000900. Similarly, the DBIL and IBIL levels were also higher in HBsAg-positive patients than in HBsAg-negative patients with genotype CC of rs17000900. However, ALB, WBC, and NEUT levels were significantly reduced in HBsAg-positive patients with genotype AC of rs17000900.

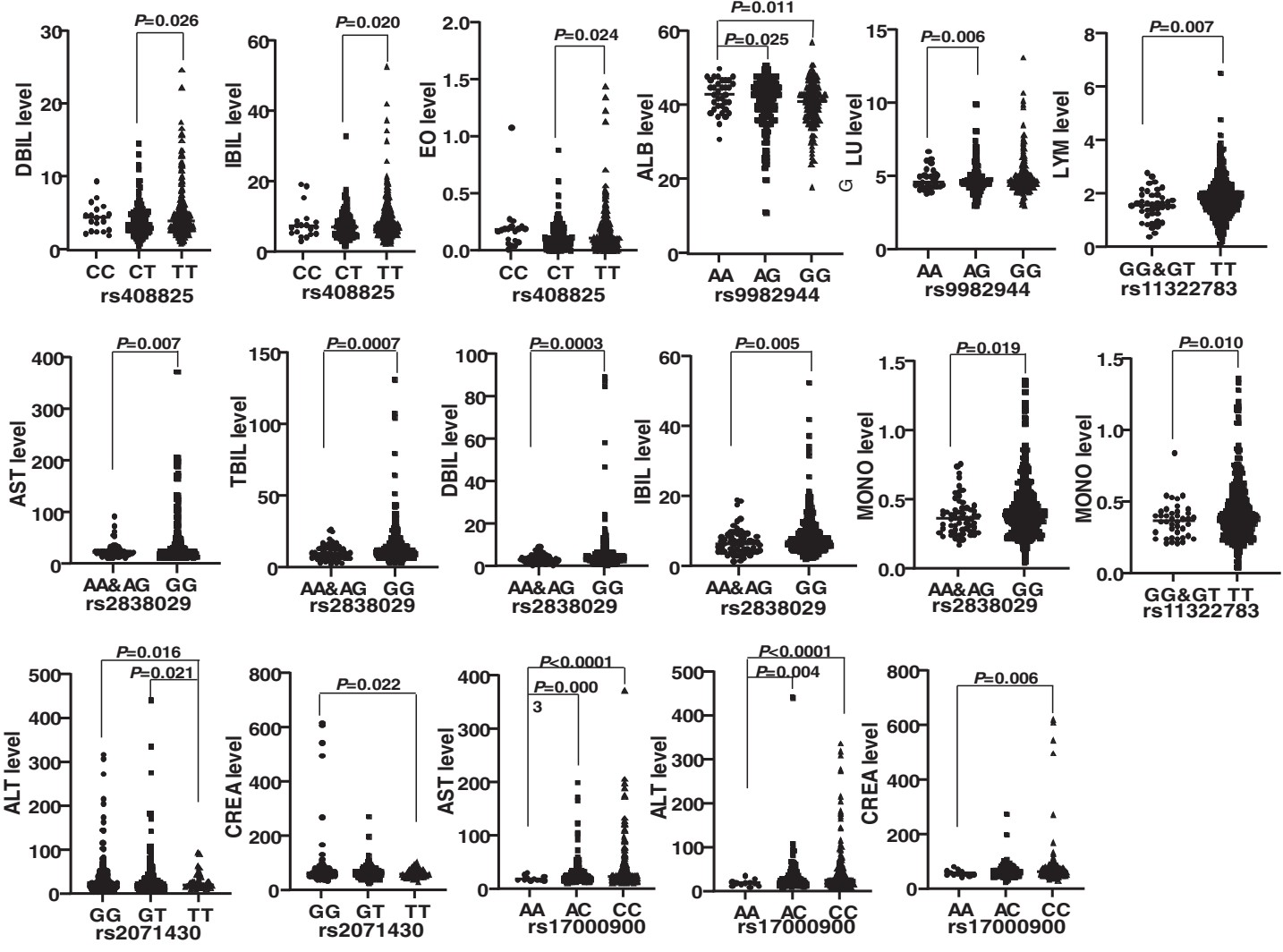

**Figure 2 Comparison of the biochemical index of HBV patients with different genotypes of each SNP.** rs11322783 is located in the *IFNL4* gene; rs2071430 and rs17000900 was in the region of *MxA* gene; rs9982944, rs408825, and rs2838029 were located in the *MxB* gene.

## DISCUSSION

A contradictory role of pegylated-interferon (Peg-IFN) was reported in HCV patients. However, it was commonly used to treat HBV and HCV infection in clinics by boosting the immune system and stimulating the expression of ISGs (*Mani & Andrisani, 2019*). HBV could interfere with the IFN signaling pathway through various mechanisms. For example, HBV core protein could inhibit the expression of the *MxA* gene stimulated by Peg-IFN (*Yu et al., 2010*). Peg-IFN treatment effect revealed significant differences among the HBV patients due to HBV genotype, viral load, and host genetic factors. The host genetic factors play essential roles in HBV infection, disease progression, and treatment effect on patients (*An et al., 2018*).

Many studies analyzed the role of genetic polymorphisms of the *IFN* genes in HBV infection (*Ben Selma et al., 2021*; *Boglione et al., 2014*). The ΔG allele of rs368234815

**Table 4 Genotype and allele frequency in patients with HBsAg-positive and -negative.**

| SNP | HBsAg-positive HBV patients (N = 210) | HBsAg-negative HBV patients (N = 238) | P-value | OR (95% CI) |
|---|---|---|---|---|
| rs11322783 | | | | |
| Genotype | | | | |
| ΔG | 0 | 1 | 0.999 | 0.000 [0.000-10.20] |
| ΔG/T | 16 | 24 | 0.455 | 0.735 [0.369–1.431] |
| TT | 194 | 213 | 0.372 | 1.423 [0.741–2.819] |
| Allele | | | | |
| ΔG | 16 | 26 | 0.313 | 0.686 [0.359–1.260] |
| T | 404 | 450 | | 1.459 [0.793–2.789] |
| rs117648444 | | | | |
| Genotype | | | | |
| AA | 0 | 0 | – | – |
| AG | 6 | 5 | 0.833 | 1.371 [0.400–3.986] |
| GG | 204 | 233 | 0.833 | 0.730 [0.251–2.503] |
| Allele | | | | |
| A | 6 | 5 | 0.835 | 1.365 [0.404–3.931] |
| G | 414 | 471 | | 0.733 [0.254–2.478 |
| rs2071430 | | | | |
| Genotype | | | | |
| GG | 110 | 120 | 0.749 | 1.082 [0.742–1.579] |
| GT | 75 | 93 | 0.525 | 0.866 [0.588–1.269] |
| TT | 25 | 25 | 0.749 | 1.151 [0.632–2.098] |
| Allele | | | | |
| G | 295 | 333 | 0.985 | 1.013 [0.760–1.355] |
| T | 125 | 143 | | 0.987 [0.738–1.316] |
| rs17000900 | | | | |
| Genotype | | | | |
| AA | 6 | 6 | 0.942 | 1.137 [0.354–3.648] |
| AC | 43 | 69 | 0.049 | 0.631 [0.407–0.976] |
| CC | 161 | 163 | 0.068 | 1.512 [0.984–2.318] |
| Allele | | | | |
| A | 55 | 81 | 0.124 | 0.735 [0.507–1.063] |
| C | 365 | 395 | | 1.361 [0.941–1.971] |
| rs9982944 | | | | |
| Genotype | | | | |
| AA | 18 | 19 | 0.957 | 1.081 [0.549–2.100] |
| AG | 94 | 110 | 0.831 | 0.943 [0.644–1.378] |
| GG | 98 | 109 | 0.929 | 1.036 [0.709–1.513] |
| Allele | | | | |
| A | 130 | 148 | 0.978 | 0.994 [0.748–1.317] |
| G | 290 | 328 | | 1.007 [0.759–1.338] |

(Continued)

| SNP | HBsAg-positive HBV patients (N = 210) | HBsAg-negative HBV patients (N = 238) | P-value | OR (95% CI) |
|---|---|---|---|---|
| **Table 4 (continued)** | | | | |
| rs408825 | | | | |
| Genotype | | | | |
| CC | 9 | 10 | 0.849 | 1.021 [0.422–2.585] |
| CT | 54 | 72 | 0.337 | 0.798 [0.529–1.217] |
| TT | 147 | 156 | 0.366 | 1.226 [0.831–1.826] |
| Allele | | | | |
| C | 72 | 92 | 0.449 | 0.864 [0.615–1.219] |
| T | 348 | 384 | | 1.158 [0.820–1.625] |
| rs2838029 | | | | |
| Genotype | | | | |
| AA | 0 | 1 | 0.950 | 0.000 [0.000–10.02] |
| AG | 25 | 35 | 0.466 | 0.784 [0.457–1.348] |
| GG | 185 | 202 | 0.393 | 1.319 [0.772–2.253] |
| Allele | | | | |
| A | 25 | 37 | 0.347 | 0.751 [0.438–1.270] |
| G | 395 | 439 | | 1.332 [0.788–2.285] |

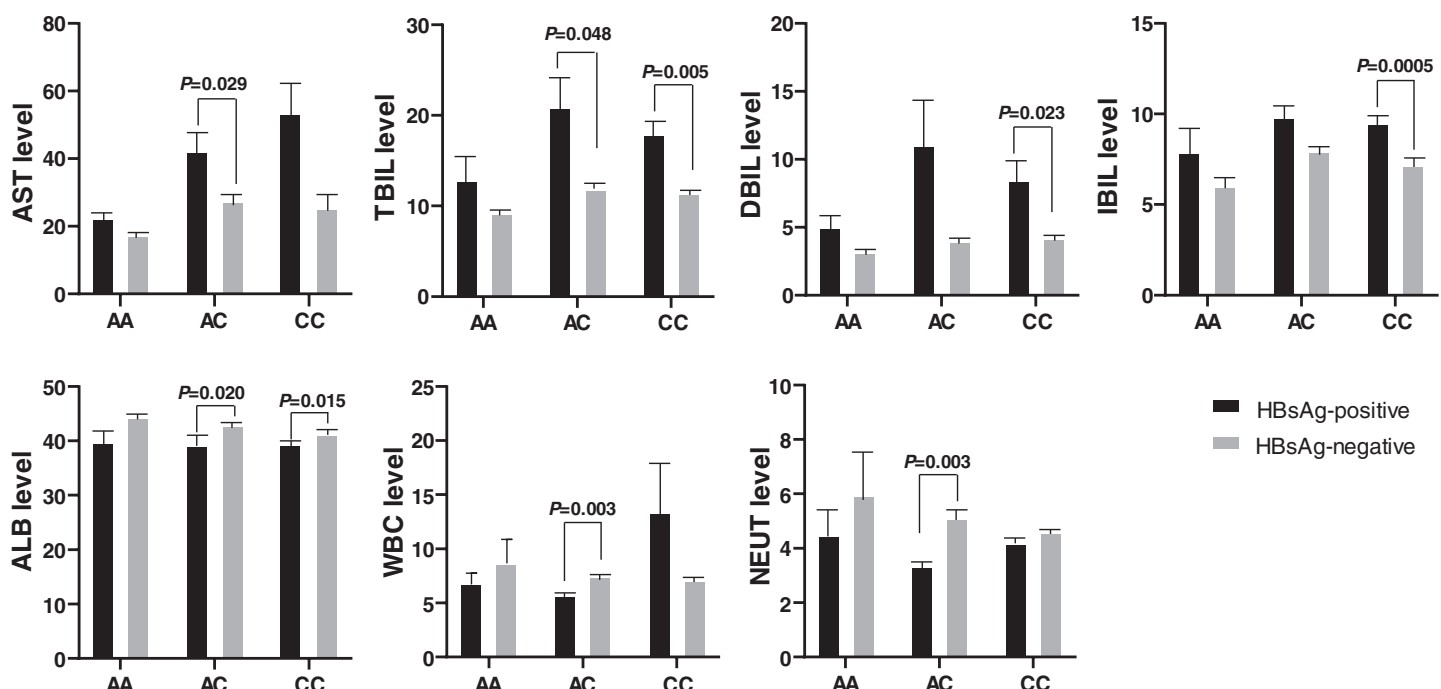

**Figure 3 Comparison of biochemical indexes between HBsAg-positive and -negative HBV patients with various genotypes of rs17000900.** The AST and TBIL level were significantly higher in patients with HBsAg-positive and genotype AC; the DBIL and IBIL level were significantly higher in patients with HBsAg-positive and genotype CC; the ALB was significantly lower in patients with HBsAg-positive and genotype AC/CC; the WBC and NEUT level were significantly lower in patients with HBsAg-positive and genotype AC.

(merged into rs11322783) was the risk factor for anti-HBs positive HBV patients, and the genotype ΔG/ΔG decreased the responsiveness of HBV vaccination (*Grzegorzewska et al., 2020*). Furthermore, the genotype of rs12979860 could modulate the HBV cccDNA levels located at the *IFNL3* and *IFNL4* genes (*Chihab et al., 2021*). In our previous study, the LYM level was much higher in HBV patients with genotype CC of rs12979860 than in patients with genotype AA (*Song et al., 2017*). In addition, the haplotype constructed by the three SNPs (rs12971396-rs8113007-rs7248668: GTA) was more frequent in HCC patients due to HBV infection than in HBV patients. However, no difference was found between HBV patients and controls (*Ma et al., 2018*). These reports suggested that genetic polymorphisms in the *IFNL4* gene could influence HBV-infected disease progressions by modulating immune reactions. However, we could not identify the association between genetic polymorphisms of the *IFNL4* gene and HBV infection in the Yunnan population.

*MxA* and *MxB* were the downstream ISGs of IFNs genes, showing direct-anti-virus function. *MxA* could repress HBV replication within transgenic mice (*Peltekian et al., 2005*). At the same time, *MxB* decreased HBV RNA levels and indirectly impaired the HBV cccDNA (*Wang et al., 2020*). However, the relationship between the genetic polymorphisms in the *MxA* and *MxB* genes and HBV infection was rarely reported. The variation –88 nt (G/T) in the promoter of the *MxA* gene expressed higher frequency in the HBV sustained patients than among non-responders (*King et al., 2002*). Although our results could not identify the association between the genotypes of SNPs in the *MxA* gene and HBV infection, the biochemical indexes of the HBV patients were influenced by genetic variations.

The *IFNL3* gene played an important role in HBV infection, viral clearance, treatment effect, and response to HBV vaccine (*Zhao et al., 2020*). Due to limited investigations, genetic variations of the *IFNL4* gene were not evident in HBV patients. The polymorphisms in the *IFNL4* gene were not associated with HBV infection and natural viral clearance in the Fan et al. study. However, a three-way interaction was identified between IFNL4/HLA-DQ and HBV infection with a multifactor dimensionality reduction test (*Fan et al., 2016*). Two SNPs rs368234815 and rs117648444 in the *IFNL4* gene, which was also analyzed in this study, could predict IFN treatment response in HBeAg-negative HBV patients (*Galmozzi et al., 2018*). Similar to our results, SNPs rs11322783 in the *IFNL4* gene could not influence the reaction of Thai HBV patients to PEG-IFN (*Limothai et al., 2015*). However, the biochemical index levels could be influenced by different genotypes of SNPs in the three genes. Thus, we suggested that SNPs in *IFNL4*, *MxA* and *MxB* genes could not directly influence HBV infection but were associated with the disease progression or treatment effect. further studies should be undertaken because there were contradictory results among various populations.

Expression of HBsAg was considered as recovered biomarker of hepatitis activity and was associated with infectious outcomes. The rs9277535 of the *HLA-DPB1* gene could develop spontaneous HBsAg seroclearance in male HBV patients (*Cheng et al., 2013*). Similarly, serum HBsAg level could reflect the interferon treatment effect (*Su et al., 2014*). In this study, genotype AC of rs17000900 was found to be the protective factor for

HBsAg-negative seroclearance among HBV patients. Moreover, biochemical indexes revealed a significant difference between HBV patients with HBsAg-positive and -negative. These results further indicated that host genetic polymorphisms were associated with HBsAg seroclearance among HBV patients.

The SNPs reported to be associated with complex traits or diseases in GWAS or linkage studies further verification. Firstly, patients should be obtained from different populations to verify the association between these SNPs and the disease. Secondly, most of these associated SNPs were located in the gene-gene locations or introns. Thus, the ability of these SNPs to tag other functional SNPs should be further identified. Moreover, SNPs in introns could influence gene expression through splicing or gene-gene interaction. Finally, these SNPs could be located within epigenetic modification region of the genes and affect function of the genes by epigenetic heritability (*Wang et al., 2011*).

## CONCLUSION

The genetic polymorphisms in the *IFNL4*, *MxA*, and *MxB* genes were not associated with HBV infection. However, they could influence the biochemical index levels of HBV patients from Yunnan.

## ACKNOWLEDGEMENTS

We thank all the participants in this study.

### Funding

This study was supported by the National Natural Science Foundation of China (32160148), the Leading Reserve Talents of Academy and Science and Technology in Yunnan Province (2019HB002), and the Yunnan Ten Thousand Talents Plan Young & Elite Talents Project. The funders had no role in study design, data collection and analysis, decision to publish, or preparation of the manuscript.

### Grant Disclosures

The following grant information was disclosed by the authors:
National Natural Science Foundation of China: 32160148.
Leading Reserve Talents of Academy and Science and Technology in Yunnan Province: 2019HB002.
Yunnan Ten Thousand Talents Plan Young & Elite Talents Project.

### Competing Interests

The authors declare that they have no competing interests.

### Author Contributions

- Kexi Zheng performed the experiments, prepared figures and/or tables, and approved the final draft.

- Yunsong Shen performed the experiments, prepared figures and/or tables, and approved the final draft.
- Xueshan Xia analyzed the data, authored or reviewed drafts of the paper, and approved the final draft.
- Yuzhu Song analyzed the data, prepared figures and/or tables, and approved the final draft.
- A-Mei Zhang conceived and designed the experiments, analyzed the data, prepared figures and/or tables, authored or reviewed drafts of the paper, and approved the final draft.

### Human Ethics

The following information was supplied relating to ethical approvals (*i.e.*, approving body and any reference numbers):

This study was approved by the Institutional Review Board of Kunming University of Science and Technology.

### Data Availability

The raw data are available in the Supplemental File.

### Supplemental Information

Supplemental information for this article can be found online at http://dx.doi.org/10.7717/peerj.13353#supplemental-information.

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
