# Peer review of "Genetic polymorphisms in the IFNL4, MxA, and MxB genes were associated with biochemical index of chronic HBV patients from Yunnan, China"

_PeerJ, doi:10.7717/peerj.13353_

## Round 0.1 · original submission · Major Revisions

Please refer to the comments from the reviewers.

Reviewer 1 ·

Basic reporting

Clear

Experimental design

Well defined

Validity of the findings

All underlying data have been provided

Additional comments

HBV infection caused hepatitis B, which was the most popular viral hepatitis in China. Interferons and its downstream interferon-stimulated geneswas important in viral infection. The authors collected HBV patients and controls to genotype genetic polymorphisms in the IFNL4, MxA, and MxB gene. Their results suggested that SNPs in these genes were associated with HBV infection in Yunnan population. Moreover, the genetic variations also influenced biochemical indexesof HBV patients. Some suggestions as following:
1. Some grammar errors could be changed, for example “All seven SNPs were tag SNP or functional SNP” should be changed to “All seven SNPs were tag SNPs or functional SNPs” in page 5.
2. In Results section, the sub-title of “Rs17000900 was associated with HBsAg seroclearance of patients” should be changed to “Rs17000900 associated with HBsAg and biochemical indexes of patients”
3. In abstract, the sentence “Above all, although the genetic polymorphisms were not ……”, which genes did these genetic polymorphisms belong to should be clearly described.

Reviewer 2 ·

Basic reporting

See below

Experimental design

See below

Validity of the findings

See below

Additional comments

Thanks for inviting me to review the manuscript entitled" Genetic polymorphisms in the IFNL4, MxA, and MxB genes were associated with biochemical index of chronic HBV patients in Yunnan, China". I have already reviewed this article in another journal (Infection, Genetics and Evolution) and my decision was reject. My comments is below:

* The abstract should be rewritten in plausible form. The name of SNPs should be added in Abstract. ALT, AST, TBIL, DBIL, IBIL, and ALB have to be fully written when mentioned for the first time.
* Inclusion and exclusion criteria are missing and are better to be added.
* Case-control study: Inappropriate study design, Control group of no considered value.
* The number of samples is small.
* It is not sufficiently clear how these particular variants were chosen for study among the numerous variants that exist in these genes. Was it merely based on previous work by others or was there some hypothesis?
* It should be discussed how a non-coding variant that does not affect protein sequence of the gene product might mechanistically affect the course of infection and response to treatment.
* Which genotypes of HBV are common in Yunnan, China? Was there relationship between these SNPs and HBV genotypes and viral load?
* The discussion has been poorly written. I would like suggest that the authors revise it, especially when they compare their results with previous studies

Reviewer 3 ·

Basic reporting

no comment

Experimental design

no comment

Validity of the findings

no comment

Additional comments

The authors collected HBV patients and controls from Yunnan Province to analyze genetic variations frequency of the IFNL4, MxA, and MxB genes. Although no association was found between these SNPs and HBV infection, the biochemical indexes levels could be influenced by the genetic variations. Moreover, genotypes of SNPs were also associated with biochemical indexes in HBsAg-positive and -negative patients. These results could provide suggestions for therapy of HBV patients.
1. The sub-title of results should be simplified.
2. Did the study use the exact same analytical systems for HBV serology and RT-PCR?
3. How about duration of patient recruitment?
4. The “Material and methods” needs to be supplemented and improved.

·

Basic reporting

The English language should be improved to ensure that an international audience can clearly understand your text. Some examples where the language could be improved are annotated in the pdf and include lines 16, 23, 41, 52, etc., There are also several grammatical errors in the manuscript as annotated in the pdf. The current phrasing makes comprehension difficult. I suggest you have a colleague who is proficient in English and familiar with the subject matter review your manuscript or contact a professional editing service.

The article needs some clarity and background information on certain things as pointed out in lines 19, 41, 56, 70, 82, 87, etc., It would also be great to have additional references to provide information to readers as pointed out in the pdf in lines 87 (reference for SnapShot method), 91 (LD), 114 (D'), etc.,

The authors conform to an acceptable format of 'standard sections' and the figures are relevant to the content of the article. However, to improve clarity of the figures, I would suggest increasing the font size of labels and uploading higher quality/resolution images. I appreciate the authors sharing raw data.

Experimental design

This research is within the Aims and Scope of the journal. Although Zheng et al., have conducted a rigorous investigation with a large study group size of 896, a clearly defined research question and knowledge gap (especially in the abstract) is missing. I suggest the authors consider stating them clearly.
Methods were described with sufficient information to be reproducible by another investigator.

Validity of the findings

I thank the authors for providing the data on which the conclusions are based, however, the conclusions need to be connected to the original question investigated. Defining the questions and gap knowledge would help make it clear.

Additional comments

The article requires significant revisions, which I feel are major enough that I would prefer to re-evaluate any revised version.

---

## Round 0.2 · Major Revisions

Please refer to the comments from reviewer 4

Reviewer 3 ·

Basic reporting

no comment

Experimental design

no comment

Validity of the findings

no comment

Additional comments

no comment

·

Basic reporting

I thank the authors for their revisions.
The figures are still of poor quality and I cannot read the text.

Experimental design

No comments

Validity of the findings

no comments

Additional comments

no comments

---

## Round 0.3 · accepted · Accept

There are no further comments.